Journal of
*Manufacturing and Materials Processing*

MDPI

*Article*

# Influence of the $CO_2$ Content in Shielding Gas on the Temperature of the Shielding Gas Nozzle during GMAW Welding

**Martin Lohse [1,*], Marcus Trautmann [1], Uwe Füssel [1] and Sascha Rose [2]**

[1] Institute of Production Engineering, Technische Universität Dresden, 01062 Dresden, Germany; marcus.trautmann@tu-dresden.de (M.T.); uwe.fuessel@tu-dresden.de (U.F.)
[2] Alexander Binzel Schweisstechnik GmbH & Co. KG, 35418 Buseck, Germany; rose@binzel-abicor.com
* Correspondence: martin.lohse@tu-dresden.de

Received: 11 November 2020; Accepted: 2 December 2020; Published: 4 December 2020

check for updates

**Abstract:** Gas metal arc welding torches are commonly chosen based on their current-carrying capacity. It is known that the current-carrying capacity of welding torches under $CO_2$ is usually higher than under argon dominated shielding gases. In this publication, the extent to which this can be attributed to the shielding gas dependent arc radiation is investigated. For this purpose, the influence of the shielding gas on the thermal load of the shielding gas nozzle of a GMAW torch was calorimetrically measured. These experiments were carried out for four different shielding gases (argon, $CO_2$, and two argon/$CO_2$ mixtures). The measurements were all performed at an average current of 300 A. The welding current was set by adjusting the wire feed rate or the voltage correction. For each case, a separate set of experiments was done. It is shown that the changed arc radiation resulting from the different shielding gases has an influence on the heat input into the gas nozzle, and thus into the torch. For the same shielding gas, this influence largely correlates with the welding voltage.

**Keywords:** GMAW; radiation; heat input; argon; $CO_2$

---

## 1. State-of-the-Art

In gas metal arc welding (GMAW), the filler material is fed via a melting wire electrode. The molten material is protected from the surrounding atmosphere by a shielding gas. In Europe, argon is mainly used as a shielding gas, with other gases such as carbon dioxide, oxygen, hydrogen, nitrogen, or helium being added, depending on the application [1,2].

The choice of the shielding gas is primarily determined by the materials to be welded [3]. Shielding gases with oxidative components, such as $CO_2$ and $O_2$, are mainly used for unalloyed and low-alloyed steels. The loss of alloy elements due to oxidation must be compensated by over-alloying the filler material [4]. The use of inert shielding gases such as argon and/or helium can prevent alloying elements from oxidizing. Pure $CO_2$ as a shielding gas is highly resistant to pore formation, but is rarely used in Europe because of the risk of carburization of the parts to be joined and the high level of spatter formation [1,3–5]. A summary of different shielding gases and shielding gas mixtures with application areas and recommendations can be found in [2].

Besides an influence on the formation of the molten bath, the shielding gas also has an influence on the arc. The drop transfer and the energy input into the electrode as well as the work piece can vary. Detailed investigations were carried out in the AiF project 17431 N [6]. For shielding gases containing argon with admixtures of max. 10% $CO_2$, the droplet transfer is uniform and mainly symmetrical to the wire axis. For $CO_2$ contents above 10%, the drop separation is increasingly asymmetrical and leads to

the formation of larger drops. Thus, the drops hit the melt pool in a larger area. For low $CO_2$ admixtures (<18%), the fusion penetration profile is finger-shaped and becomes lenticular with increasing $CO_2$ content, whereby the fusion penetration depth and seam height decrease. With increasing $CO_2$ content, the arc length continues to decrease.

The influence of the shielding gas on the torch components has not been scientifically investigated so far. The design of welding torches is based on the current carrying capacity as a function of the duty cycle, which is determined, e.g., according to DIN EN 60974-7. This specifies argon for MIG welding of aluminium and argon-$CO_2$ mixed gases with 15 to 25% $CO_2$ content for MAG welding of mild steel. The resulting permissible currents for $CO_2$ at a constant duty cycle are specified by the manufacturers as approximately 30–50 A (10–20%) higher for $CO_2$ than for argon [7,8]. These specifications apply to both gas-cooled and liquid-cooled torches. This circumstance is usually explained with the increased cooling effect of $CO_2$, whereas no proof is provided. There are also no studies on this topic in the literature. Only in the work of Haelsig [9], it is determined by calorimetric measurements on a GMAW process that approximately 7–9% of the electrically supplied power of the welding process is dissipated by cooling the torch. Depending on the process, this can be up to a Kilowatt and explains the need for water-cooling in welding torches under high loads. However, no publication could be found on how large the contribution of heat radiation is, especially for different shielding gases.

Nevertheless, there are publications on the temperature-dependent material data of different shielding gases, which were calculated assuming the local thermodynamic equilibrium. Data can be found, for example, in [10] for argon and in [11] for $CO_2$. Selected values are shown in Figure 1. In the area of the shielding gas nozzle, temperatures of approximately 300–900 K occur. In this temperature range, both the specific heat capacity and the thermal conductivity of $CO_2$ are in part significantly higher than those of argon. This leads to a higher cooling effect of $CO_2$ compared with argon in this temperature range. However, no quantification of this effect can be derived from the material data alone, as the formation of the gas flow also has an influence on the heat transfer. The density and viscosity of the gases also play a role. In the temperature range of 300–900 K, the viscosity of argon is about double as high as for $CO_2$, while the density of $CO_2$ is about 10% higher than that of argon.

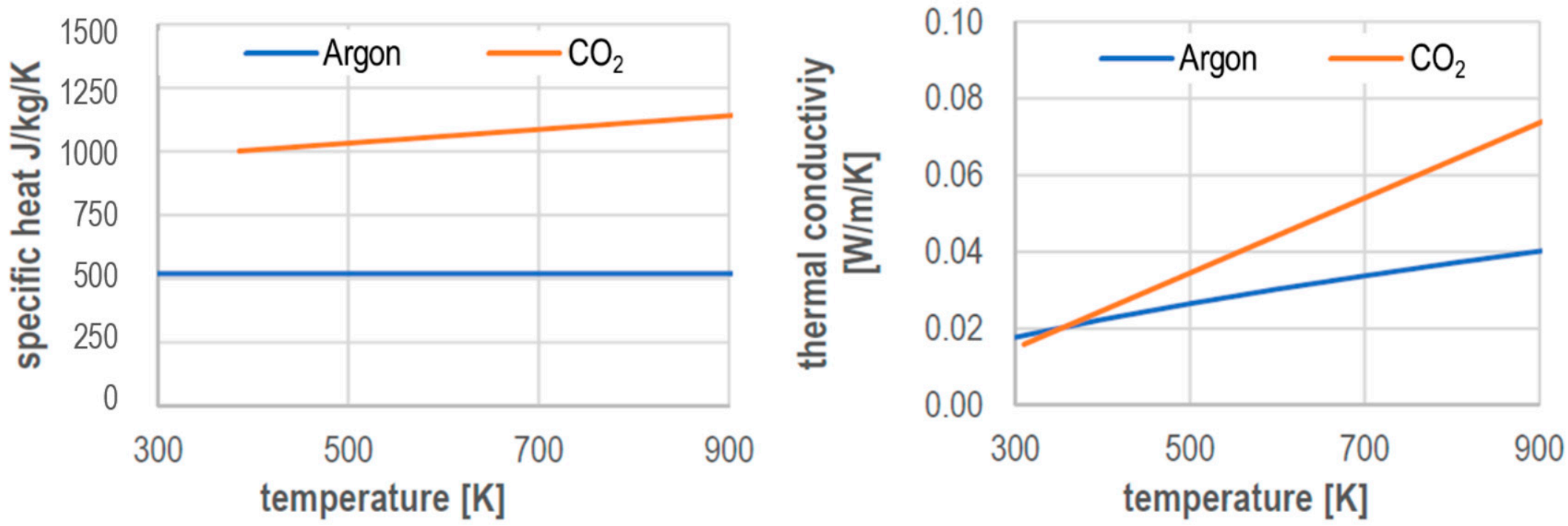


**Figure 1.** Specific heat and thermal conductivity of argon and carbon dioxide in comparison [10,11].

The arc temperatures and thus the radiated heat also differ between the shielding gases. Corresponding temperatures were published in [12,13] and are shown in Figure 2. In [12], temperatures of approximately 9000–10,000 K in the metal-vapor dominated arc core were determined for an arc in an argon shielding gas; see Figure 2. Outside this range, the temperatures in the plasma were approximately 12,000 K. In [13], shielding gas mixtures of $CO_2$ with high argon content and pure $CO_2$ were investigated. It is shown that the minimum temperature in the arc core is more pronounced for argon-rich shielding gas mixtures than under pure $CO_2$; see Figure 2. The determined arc temperatures for an argon mixture with 18% $CO_2$ were about 10,000 K in the arc core and between 11,000 K and almost 14,000 K in the plasma. The temperatures for an arc under pure $CO_2$ shielding gas, however, are comparatively lower at approximately 8000—9000 K.

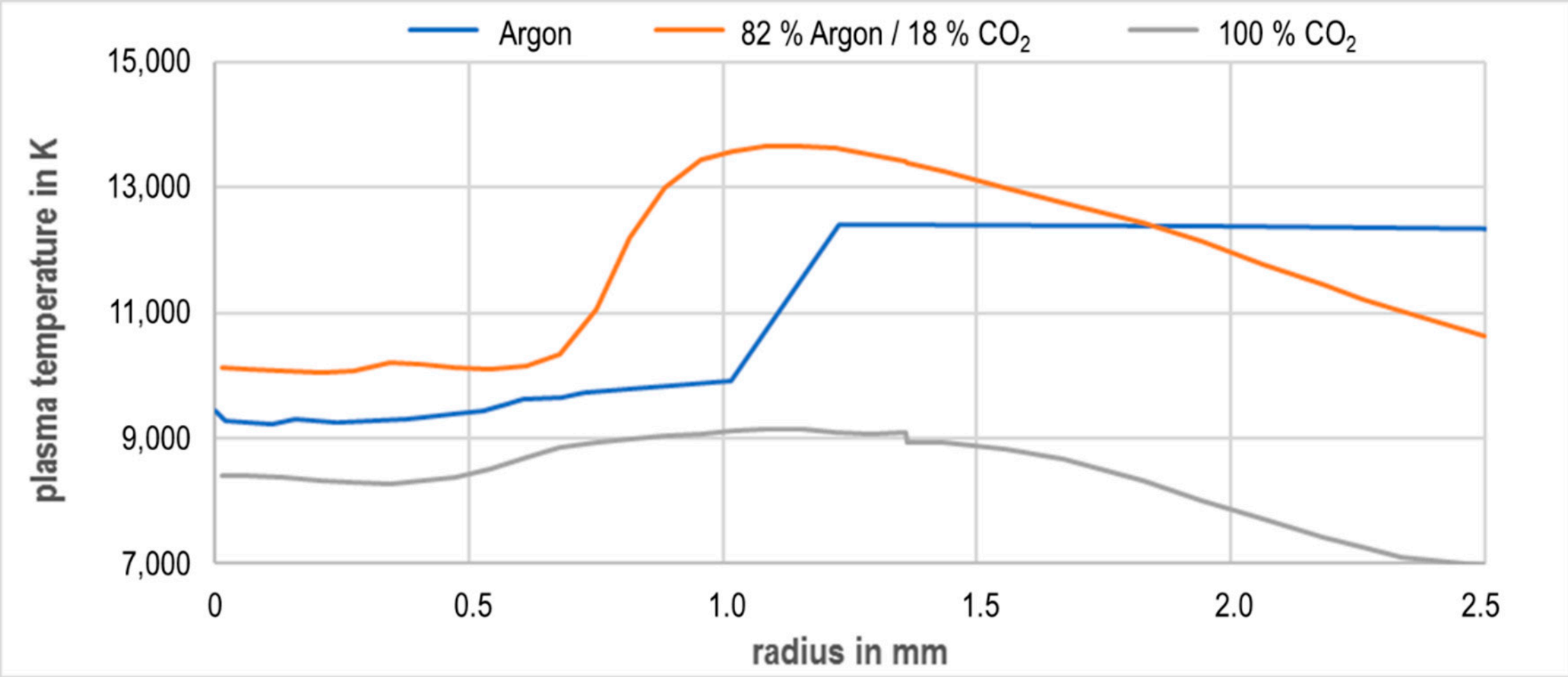


**Figure 2.** Measured arc temperatures for a gas metal arc welding (GMAW) pulse process. Values under argon determined in [12] for an arc 1040 ms after the start of the high current phase and 1.7 mm above the workpiece. Values for 82% argon/18% $CO_2$ and $CO_2$ determined in [13] for an arc 980 ms after the start of the high current phase approximately 1.5 mm above the workpiece. The welding current at the time of measurement was about 420 A for argon, about 250 A for 82% argon/18% $CO_2$, and about 330 A for $CO_2$.

The radiation behaviour can be described by the net emission coefficient (NEC). In [14,15], corresponding values for argon and $CO_2$ in the temperature range from 300 K to 30,000 K are published and they are shown in Figure 3.

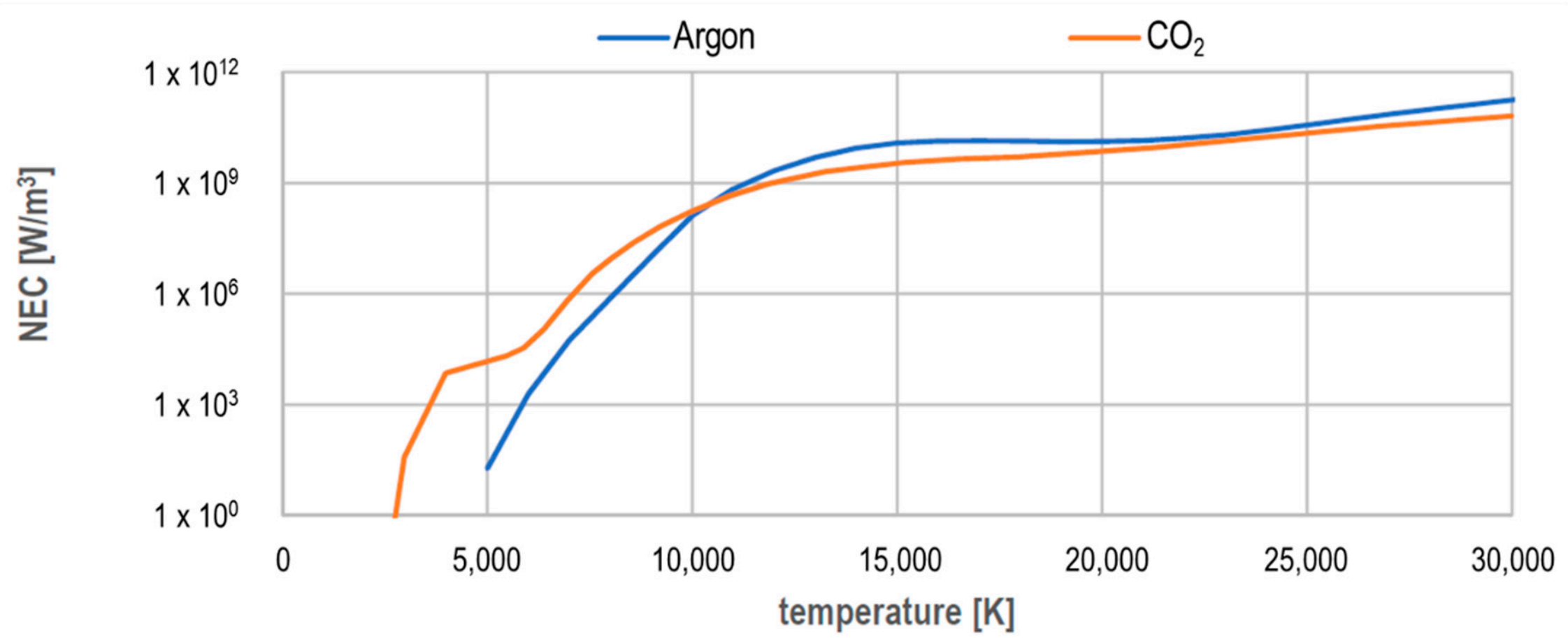


**Figure 3.** Net emission coefficient of argon and carbon dioxide compared. Data is taken from [14,15].

From the published temperatures, it follows that the NEC for an arc under argon is between $1.0 \times 10^7$ W/m$^3$ and $3.5 \times 10^9$ W/m$^3$. For a $CO_2$ arc, the NEC decreases to values of $9.0 \times 10^6$ W/m$^3$ and $6.7 \times 10^7$ W/m$^3$. When $CO_2$ is used as a shielding gas, this results in lower heat radiation, and thus a lower thermal load on the torch components due to the arc radiation. However, higher arc temperatures were measured for a shielding gas mixture of 82% argon and 18% $CO_2$. For these values, the NEC is between $1.8 \times 10^8$ W/m$^3$ and $5.3 \times 10^9$ W/m$^3$, so that a higher radiation load of the torch components could be concluded. Based on the material data alone, the heat input into the torch components cannot be quantified, as other parameters, such as the radiant volume, i.e., the arc size, must be known.

## 2. Aim of the Study

According to the state-of-the-art, GMAW torches, which are designed according to DIN EN 60974-7, can generally be loaded with a higher current under a $CO_2$ shielding gas than under argon. The published material data of the shielding gases and the temperatures in the arc already give a possible explanation for this "cooling effect". However, no quantification of individual effects of the heat transfer from the arc and shielding gas to the torch components has yet been made.

The heat input into a GMAW welding torch is determined by convection, solid state radiation, and arc radiation. In this work, the heat, which is introduced into the shielding gas nozzle by the arc radiation, is determined calorimetrically. It is assumed that the radiation influence on other torch components exhibits analogous behaviour. For the calorimetric measurement, the shielding gas nozzle is cooled down to almost room temperature. Thus, a convective heat transfer can be omitted.

## 3. Experimental Setup

To determine the radiation input by the arc in an isolated manner, a heavily cooled shielding gas nozzle is used, which absorbs the majority of the arc radiation in the direction of the torch. Figure 4 shows on the left side the principle of the heat sink, which serves as a shielding gas nozzle. The heat sink consists of a round copper piece with an internal cooling channel for water-cooling. On the bottom side, a shortened gas nozzle is screwed in. The heat dissipated by the cooling water is determined calorimetrically. Because the heat sink is cooled down to almost room temperature and is insulated against the torch, the cooling capacity corresponds to the heat input by radiation.

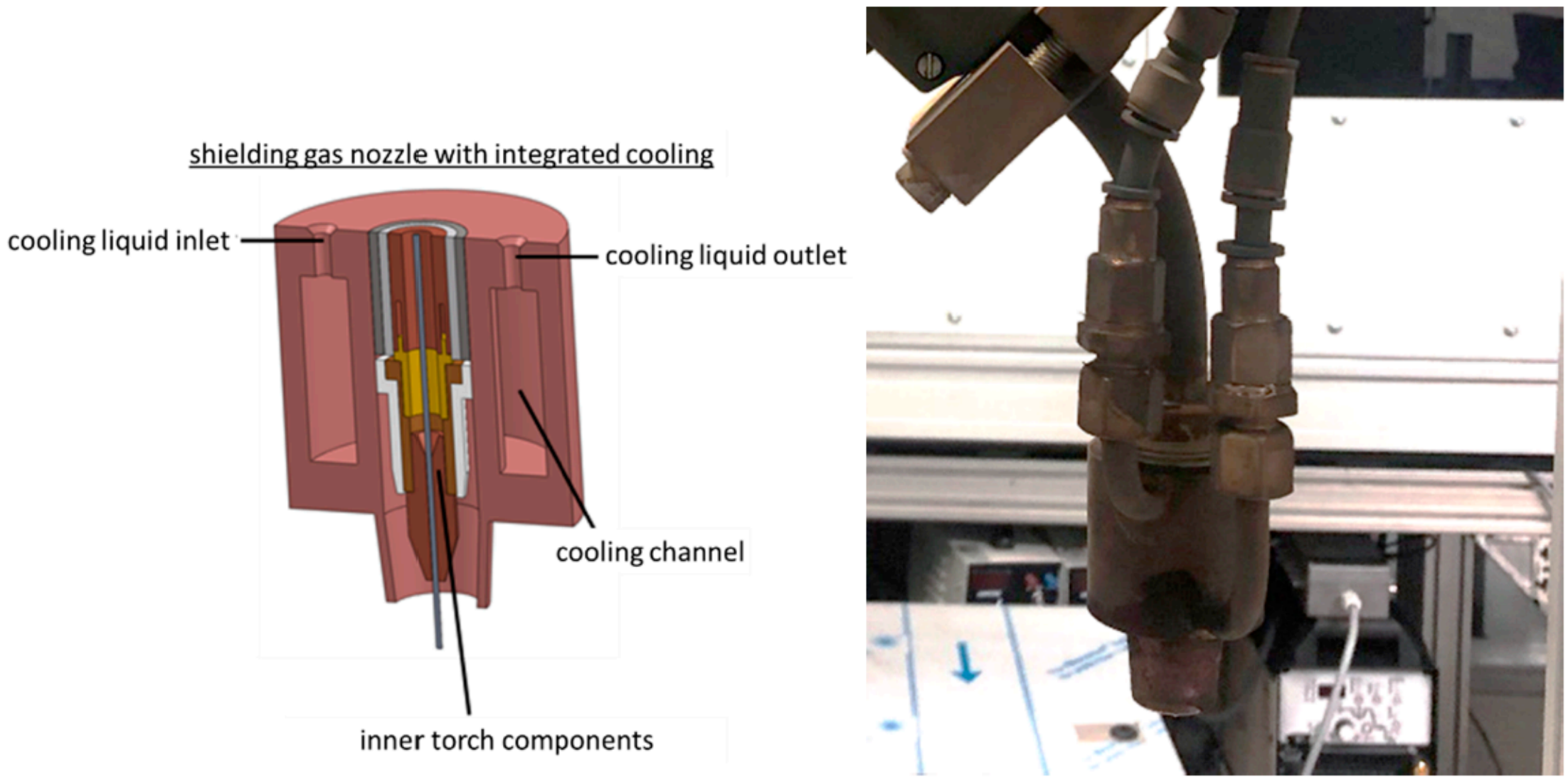


**Figure 4.** Design of the gas nozzle cooling and current-carrying torch parts. (**Left**) principle sketch; (**Right**) design of the torch.

In Figure 4, on the right side, the heat sink is mounted on a liquid cooled 500 A GMAW torch from ABICOR Binzel, which in turn is mounted on a three-axis gantry. For the experiments, a power source from EWM "Phoenix 521 Expert Puls forceArc" was used.

The process was recorded with the help of a single-lens reflex (SLR) camera and a high-speed camera in order to assess the visible part of the arc in a qualitative manner. Furthermore, current and voltage were recorded with a sampling rate of 10 kHz. A filter with 810 nm was used for the recordings with the SLR camera. For the recordings with the high-speed camera, a combination of an 810 nm and an 808 nm filter was used.

For all experiments, a constant welding current of 300 A and a constant contact tip distance of 17 mm were set. These parameters were chosen following the AiF project [6], in order to be able to run

the same welding current for all shielding gases investigated. The welding current was adjusted via the wire feed rate or voltage correction. This results in two measurement series:

- Measurement series 1: adjusted wire feed (constant voltage correction);
- Measurement series 2: adjusted voltage correction (constant wire feed).

## 4. Heat Input with Adjusted Wire Feed Rate

Table 1 below lists the process parameters for the experiments. In order to set a current of 300 A for the various shielding gases, a characteristic curve for the shielding gas was set at the power source by selecting the appropriate job. The welding current was determined by direct measurement using a Dewetron measuring device and averaged over a time of approximately 5 s. The wire feed rate was then adjusted until an average current of approximately 300 A could be measured.

**Table 1.** Process parameters of the experiments.

| Welding Position According to DIN EN ISO 6947 | | Flat Position |
|---|---|---|
| type of welding | | Overlap joint |
| droplet transfer | | mainly free of short circuits (determined short circuit time in the results) |
| wire diameter | [mm] | 1.2 |
| welding current | [A] | 300 |
| voltage correction | [V] | 1 |
| wire feed | [m/min] | variable |
| wire material | | G3Si1/DIN 8559: SG2/AWS: A5.18 |
| power supply | | direct current, not pulsed, wire positive |
| contact tip distance | [mm] | 7 |
| welding speed | [mm/s] | 12 |
| welding time | [min] | ~4 |
| weld seam length | [m] | ~3 |
| shielding gas composition | | 100% Argon<br>Argon 97.5% $CO_2$ 2.5%<br>Argon 82% $CO_2$ 18%<br>100% $CO_2$ |
| flow rate of the shielding gas | [l/min] | 20 |
| flow rate of cooling water for calorimetry | [l/min] | 1.7 |

In order to minimize the influence of process fluctuations and of the heating of the torch on the results, a seam length of 3 m is welded. For this purpose, the torch was moved over the work piece in a meandering manner. The welding speed was 12 mm/s and the length of one meander path 600 mm. The distance between the individual paths was 15 mm. Five connecting paths were welded, resulting in a welding time of approximately 4 min.

### 4.1. Parameter

Table 2 summarizes the determined wire feed rates. Furthermore, the measured current, voltage, and calculated short circuit time, as well as electrical power, are shown.

**Table 2.** Process parameters for the individual shielding gases.

| Shielding Gas | Argon 100% $CO_2$ 0% | Argon 97.5% $CO_2$ 2.5% | Argon 82% $CO_2$ 18% | Argon 0% $CO_2$ 100% |
|---|---|---|---|---|
| Wire feed [m/min] | 7.6 | 9.0 | 9.0 | 10.5 |
| Short circuit time [%] | 0 | 0 | 0 | 10 |
| average current [A] | 300 | 301 | 302 | 304 |
| average voltage [V] | 30.18 | 27.78 | 31.25 | 26.96 |
| average power [W] | 9050 | 8360 | 9440 | 8200 |

### *4.2. Current and Voltage Measurement*

Figure 5 shows the current and voltage curves of the welding experiments. From the curves of the voltage, it can be seen that the drop transition is short-circuit free for all measurements except for 100% $CO_2$.

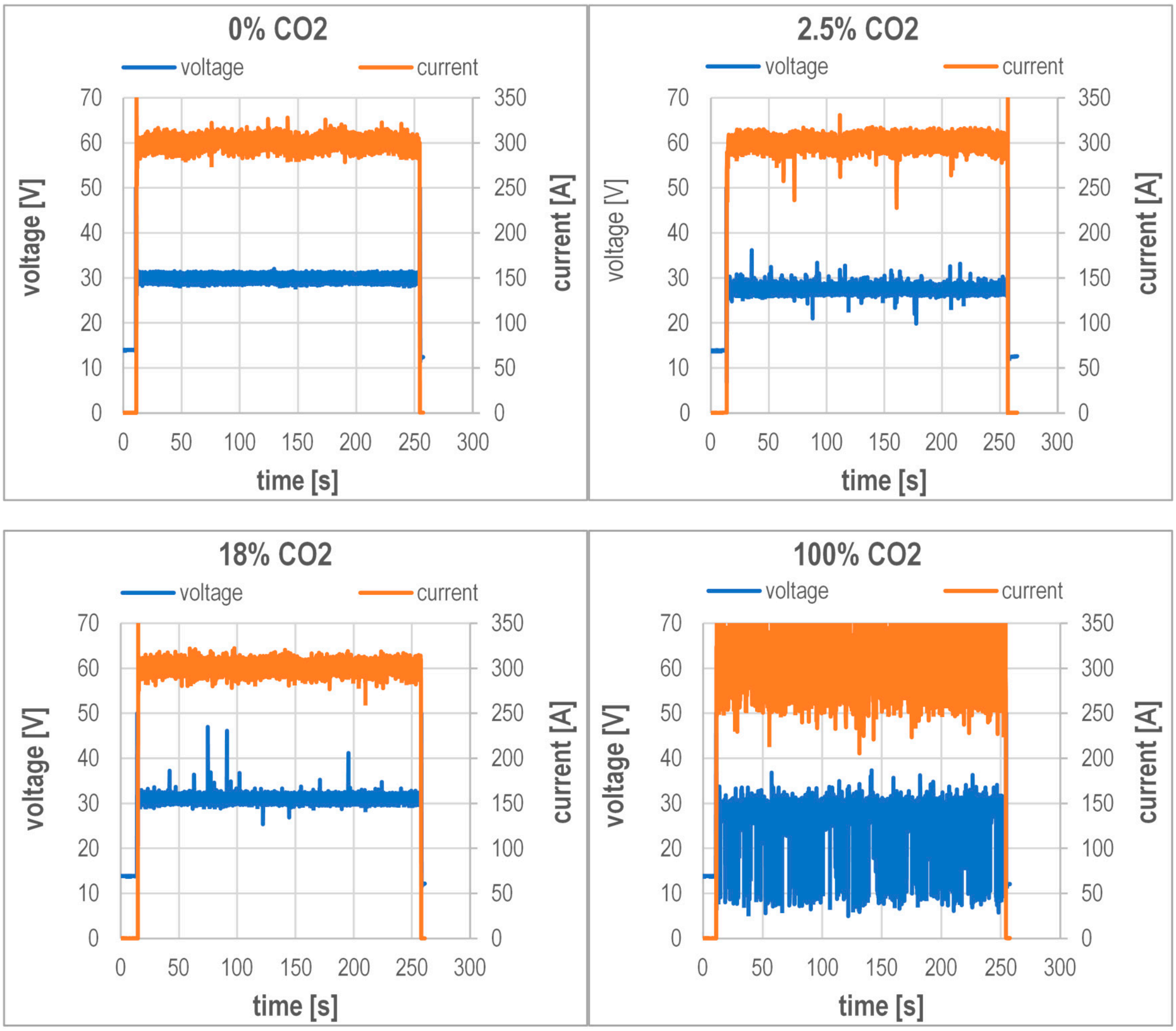


**Figure 5.** Current and voltage for an argon shielding gas with 0%, 2.5%, 18%, and 100% $CO_2$.

Figure 5 shows that the scatter of the measured values increases with increasing $CO_2$ content in the shielding gas. At the same time, the visible part of the arc shortens, as shown in Figure 6. From the electrical signals, a short-circuit proportion of approximately 10% results for 100% $CO_2$, while the process runs without a short circuit for the other gases. The photos shown in Figure 6 illustrate the visible arc length and were taken with the SLR camera and an 810 nm filter.

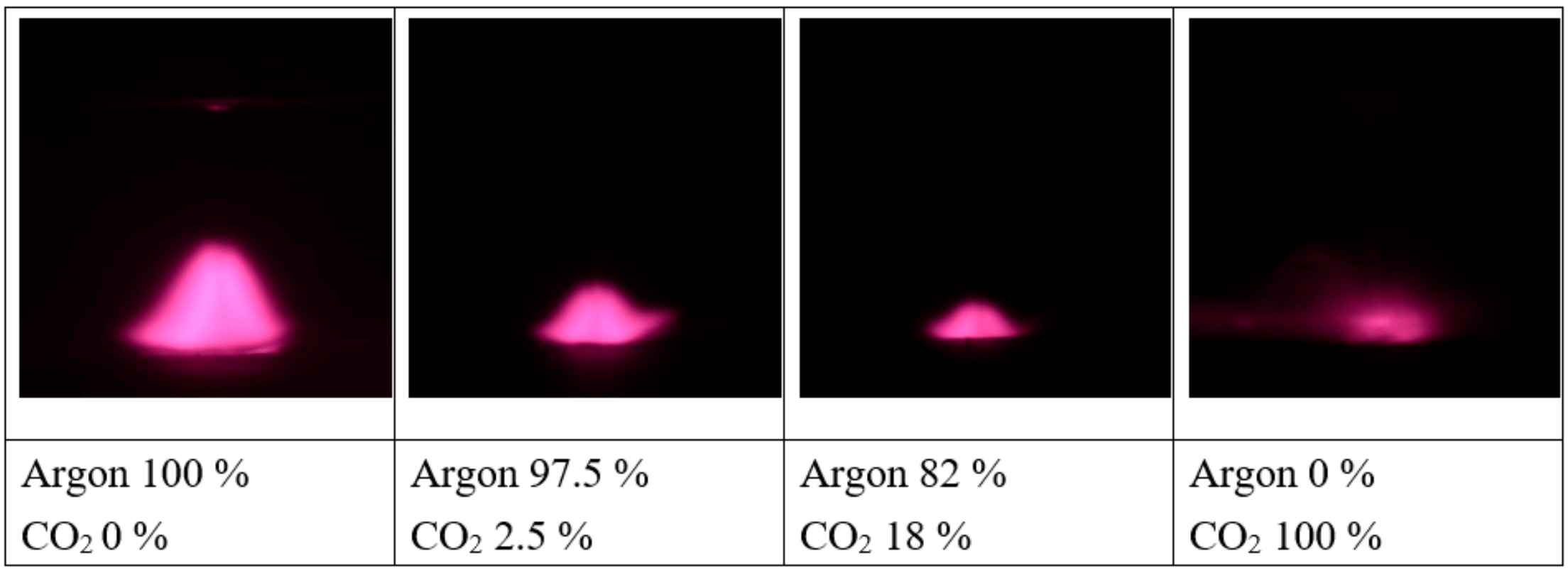


**Figure 6.** Visible arc length of the examined shielding gases in qualitative comparison.

### 4.3. Heat Input Measurement

Figure 7 shows the heat input into the gas nozzle over the measuring time calculated from the measuring data.

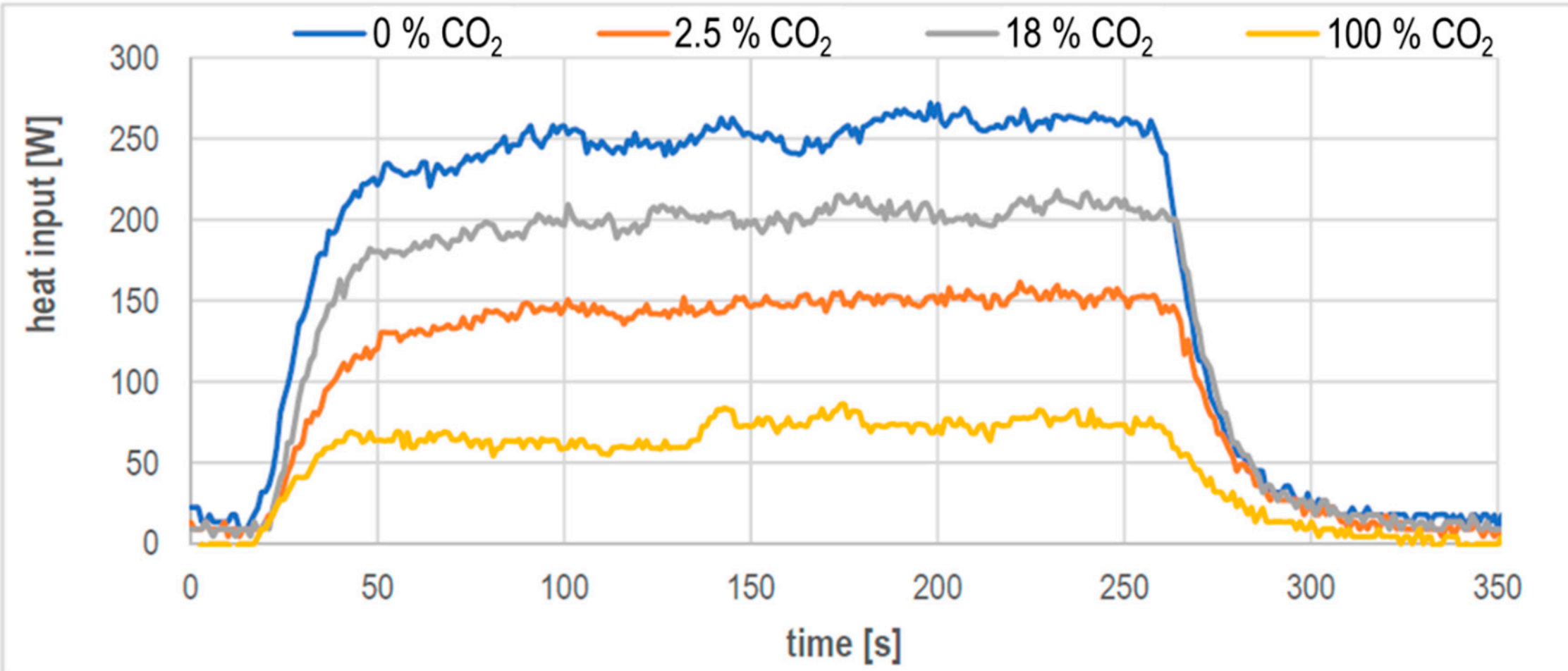


**Figure 7.** Heat input into the gas nozzle for the examined shielding gases.

For the calculation of the average heat input, measured values between 200 and 250 s are selected, as steady state measurement can be assumed.

In Figure 8, the corresponding values including their variance over the considered period are shown in a diagram.

From the results, it can be seen that the heat input is clearly dependent on the choice of shielding gas. By adding $CO_2$ to the shielding gas argon, the heat input introduced into the gas nozzle by radiation is reduced. This reduction is the least pronounced at a $CO_2$ content of 18%. For a $CO_2$ content of 2.5%, the reduction is already 40% and, for 100% $CO_2$, 70%. Taking into account the time in which the welding process is in short circuit, and thus does not emit any radiation, it can be estimated that the heat input at 100% $CO_2$ is approximately 10% higher, resulting in a value of approximately 81 W. This difference is small enough for the statements derived so far to stay valid.

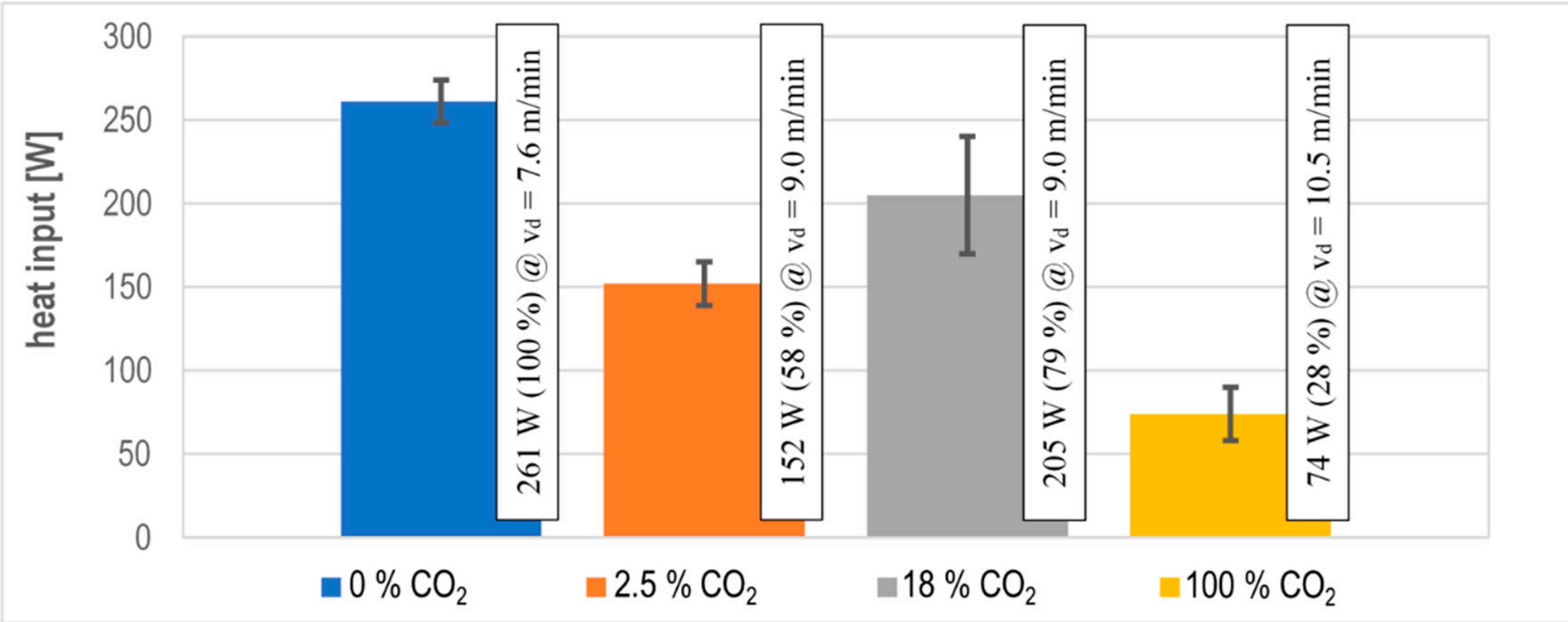


**Figure 8.** Average heat input into the gas nozzle for the examined shielding gases with variance over the evaluated period at 300 A.

### 4.4. Interim Summary

In this series of measurements, the influence of the shielding gas on the heat input by arc radiation into the gas nozzle was investigated. The admixture of $CO_2$ in argon leads to a reduction of the heat radiation into the torch components by up to 40%. The use of 100% $CO_2$ leads to a reduction of 70%.

Figure 8 shows a local maximum in heat input at 18% $CO_2$. This result is interesting, because Figure 6 shows that the arc length decreases continuously with the increasing $CO_2$ content. Consequently, the radiant volume also decreases steadily. From the publications on arc temperature, however, it is known that the temperature in the arc is higher for 18% $CO_2$ than for argon or $CO_2$; see Figure 2. Consequently, this effect can reduce the influence of the reduced arc length, and thus lead to the local maximum.

## 5. Heat Input with Constant Wire Feed Rate

From the experiments carried out so far, it can be seen that the GMAW process has very different arc lengths for different shielding gases at constant current and adjusted wire feed. Therefore, the measured heat input into the shielding gas nozzle cannot be clearly assigned to the influence of the shielding gas. As a result, the tests are performed with constant arc length. The procedure from the AiF project [6] was applied.

For the test, the same characteristic curve is used for all shielding gases, which corresponds to the same set job at the power source, and the same wire feed is used. Then, the voltage correction is set so that the same welding current results for each shielding gas. This means that the same deposition rate is set for all tests and, consequently, the arc length should be the same.

### 5.1. Parameter

Table 3 summarizes the determined voltage corrections for the examined shielding gases and the adjusted wire feed. Furthermore, the measured current and voltage and the calculated short circuit time and power are shown. With these values, the deviation of the average welding current from the targeted 300 A was the smallest.

**Table 3.** Process parameters for the individual shielding gases.

| Shielding Gas | Argon 100% $CO_2$ 0% | Argon 97.5% $CO_2$ 2.5% | Argon 82% $CO_2$ 18% | Argon 0% $CO_2$ 100% |
|---|---|---|---|---|
| Voltage correction [V] | −4 | −2.5 | 4.9 | 14.9 |
| Wire feed [m/min] | 8.3 | 8.3 | 8.3 | 8.3 |
| Short circuit time [%] | 0 | 0 | 0 | 0 |
| average current [A] | 302 | 303 | 303 | 286 |
| average voltage [V] | 28.15 | 29.53 | 36.92 | 46.24 |
| average power [W] | 8500 | 8950 | 11,190 | 13,220 |

## 5.2. Current and Voltage Measurement

Figures 9 and 10 show the current and voltage curve of the welding experiments. The welding process is short-circuit-free for all measurements.

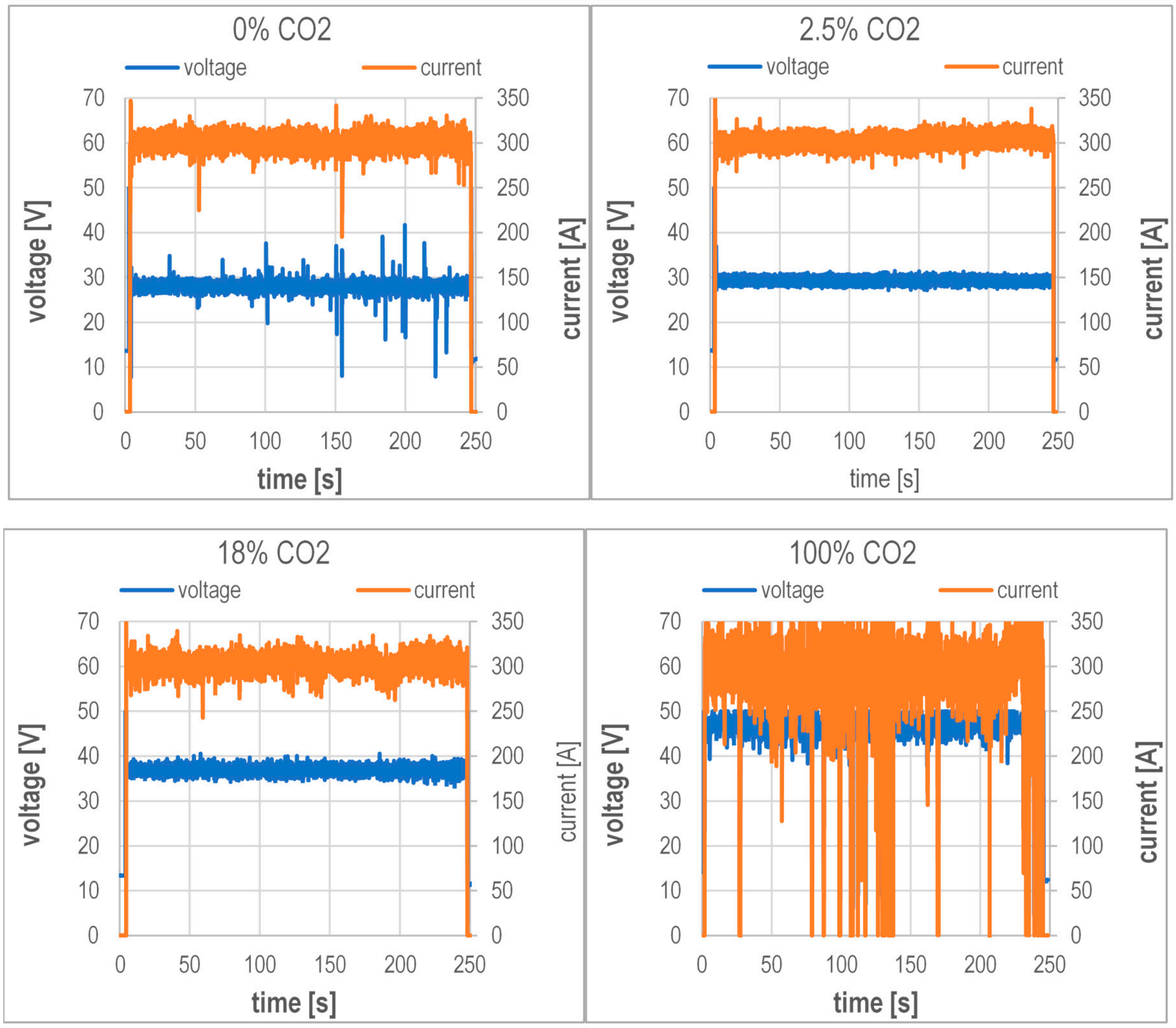


**Figure 9.** Current and voltage for an argon shield gas with 0%, 2.5%, 18%, and 100% $CO_2$ content.

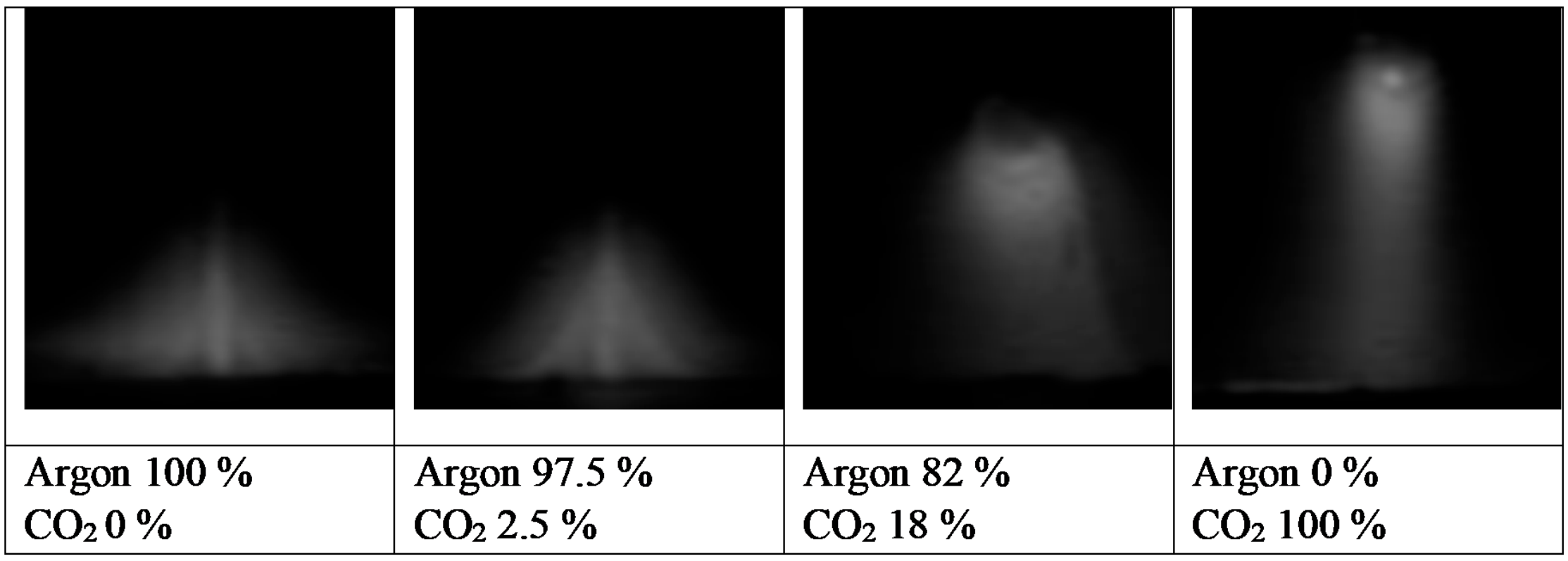


**Figure 10.** Visible arc length of the examined shielding gases in qualitative comparison.

It can be seen from Figure 9 that the scatter of the measured values increases with the increasing $CO_2$ content in the shielding gas. At the same time, the visible part of the arc lengthens, as shown in Figure 11. Starting from an admixture of 18% $CO_2$, the arc cone is visibly enlarged.

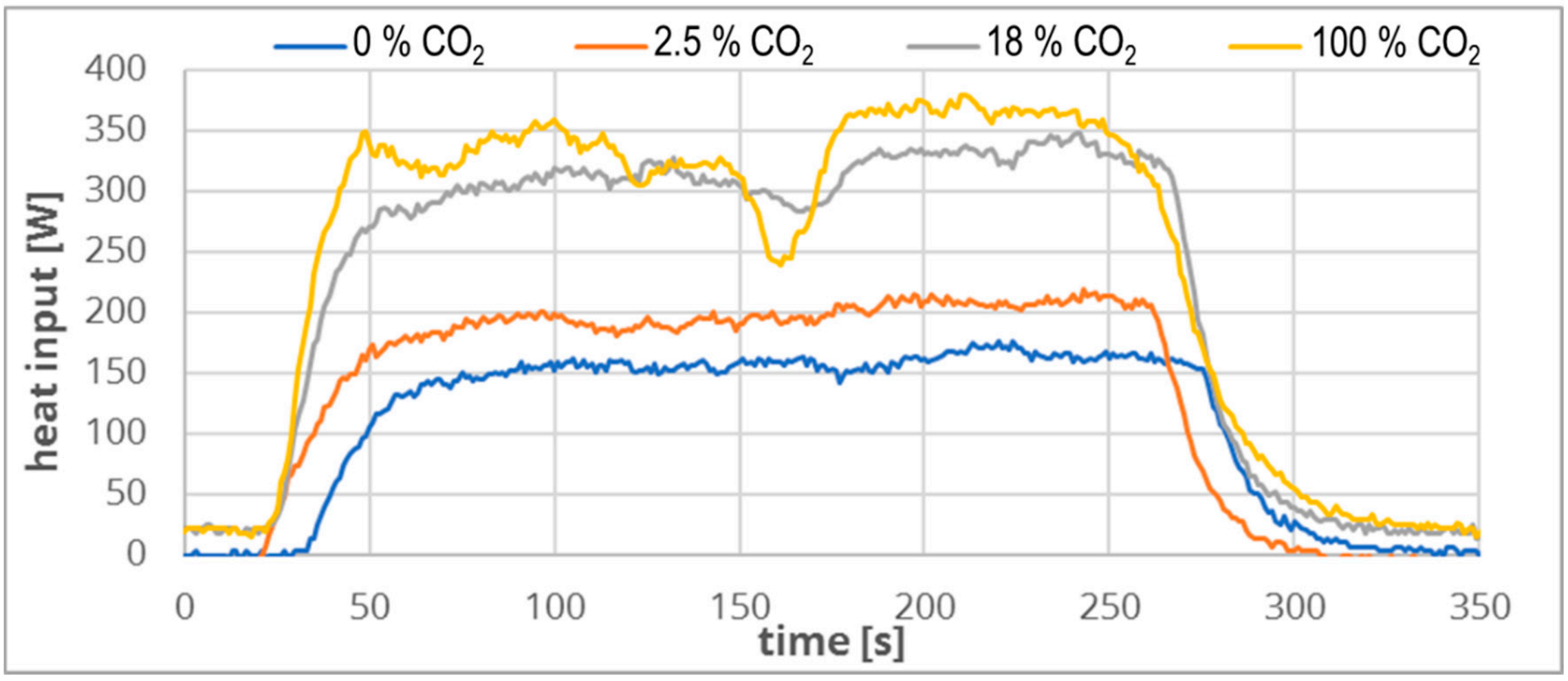


**Figure 11.** Heat input into the gas nozzle for the examined shielding gases.

The photos shown in Figure 10 were taken with a high-speed camera and a combination of an 808 nm and an 810 nm filter.

Similar to the results of the AiF project [6], the same arc length was expected for the different shielding gases owing to the constant current and constant wire feed. In the respective project, the process is illuminated with a laser and recorded with the help of a filter that only allows the laser radiation to pass. Thus, it could be proven that the distance of the solidus line on the welding wire to the surface of the work piece is approximately the same for constant current and wire feed values. Figure 10 shows that at least the visible part of the arc is nevertheless of a different length.

### *5.3. Heat Input Measurement*

Figure 11 shows the heat input into the gas nozzle over the measuring time.

For all measurements, it can be seen that the heat input has settled after 100 s. For the measurements with 18% or 100% $CO_2$, a medium to strong decrease of the energy input can be seen in the middle of the measurement. This was attributed to a thermal deformation of the sheets. As soon as the deformed area is left, the energy input rises again to a constant average value. For the calculation of the

average heat input, measuring values between 200 and 250 s are chosen, because here, steady values are available for all measurements.

In Figure 12, the corresponding mean values including their variance over the observed period are shown in a diagram.

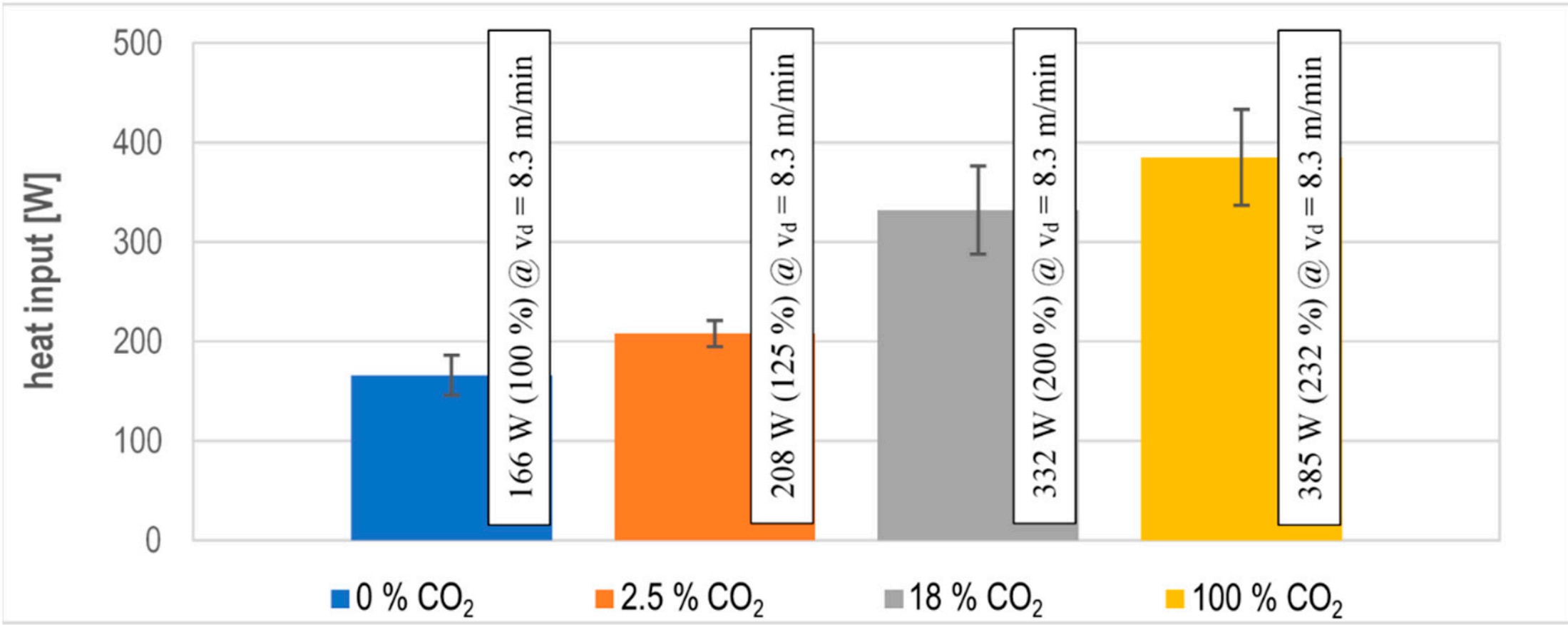


**Figure 12.** Average heat input into the gas nozzle for the examined shielding gases with variance over the evaluated period.

Figure 12 shows that, in contrast to the previous investigations, the heat input increases with the increasing $CO_2$ content. There is also no local maximum at a content of 18% $CO_2$ in the shielding gas.

Figure 13 shows the results of both test series in a diagram. In addition, the measured mean current and the measured mean voltage are shown at each entry. From this comparison, it can be seen that the radiation-induced energy input can vary considerably even with the same shielding gas. Thus, the largest and smallest measured heat input was determined for 100% $CO_2$. Therefore, the radiation-related heat input into the shielding gas nozzle can be increased or reduced for the same shielding gas by adjusting the process voltage.

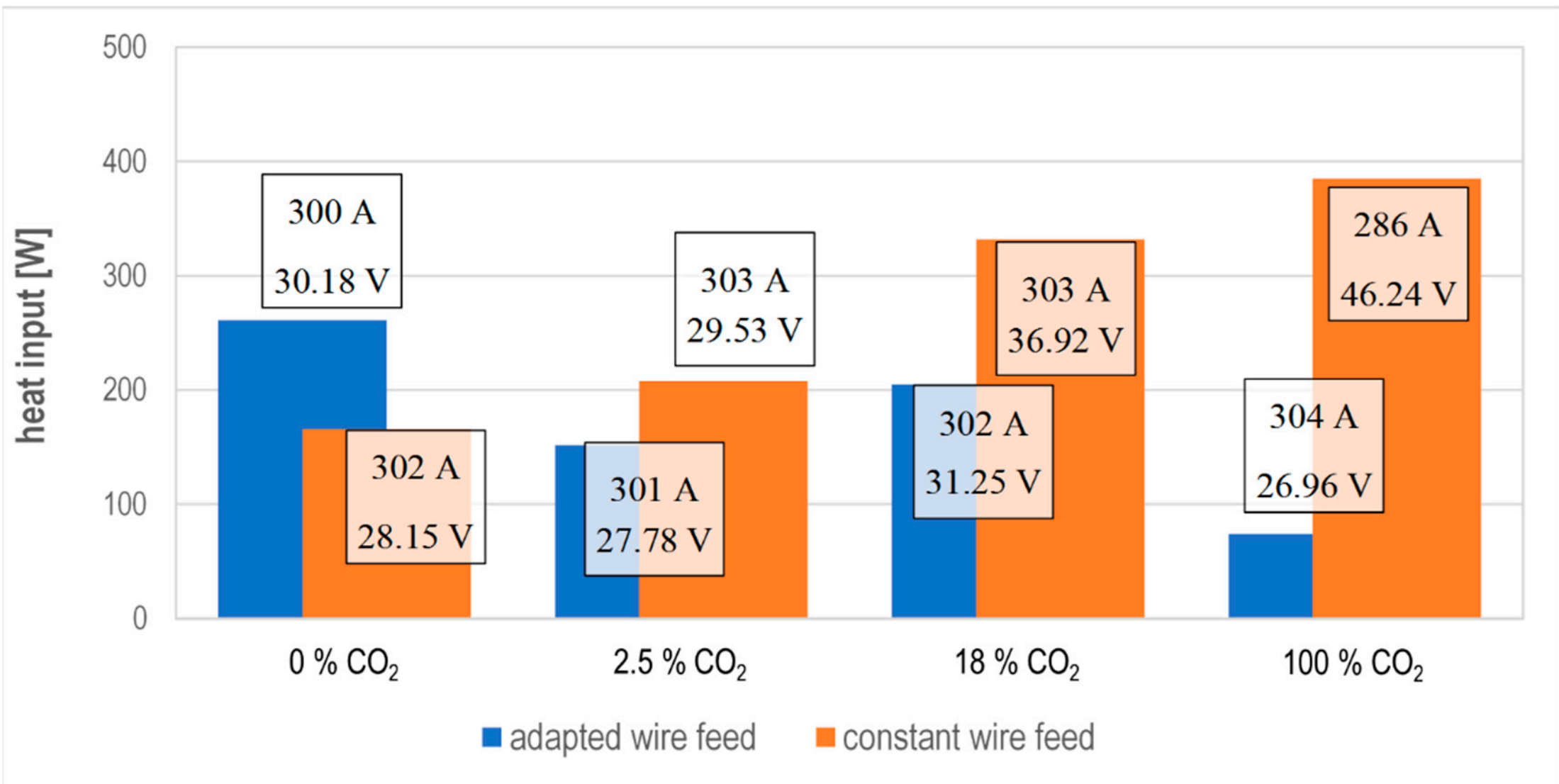


**Figure 13.** Average heat input into the gas nozzle for the examined shielding gases from the two test series.

## 6. Summary

With the measurements carried out, the influence of the shielding gas in GMAW welding on the energy input into the shielding gas nozzle due to radiation was quantified. For the evaluation of the results, it has to be considered that the different process gases partly differ strongly in their process voltage and arc characteristics.

In the first part of the experiments, the process parameters were selected by adjusting the characteristic curve and the wire feed to the shielding gas. These tests showed a reduction of the radiation-induced energy input into the shielding gas nozzle for increasing $CO_2$ contents, whereby a local maximum exists for a mixture of 82% argon and 18% $CO_2$. At the same time, a decrease of the visible arc length and, consequently, of the radiant volume was observed with the increasing $CO_2$ content. This decrease is presumably counteracted by higher temperatures in the arc at 18% $CO_2$ compared with 100% argon or 100% $CO_2$.

In the second part, the process parameters were selected with the aim of achieving a constant radiating volume. Therefore, the selection of parameters was analogous to the results from [6]. However, the investigations in this publication show that the visible arc length for increasing $CO_2$ contents in the shielding gas nevertheless increases steadily. The experiments also show that an increase in the $CO_2$ content in the shielding gas is accompanied by an increase in the radiation-related energy input into the shielding gas nozzle.

In the tests carried out, the smallest as well as the largest radiation-related energy input into the shielding gas nozzle was determined for an arc below 100% $CO_2$. This effect was achieved with a massive increase of the welding voltage. This also increased the arc length. However, this correlation does not allow for a causation. On the contrary, under any shielding gas, the radiation-related energy input into the shielding gas nozzle can be increased by increasing the voltage.

Using standard parameters, it can be assumed that an arc under $CO_2$ emits less radiation into the shielding gas nozzle than an arc under argon–$CO_2$ mixtures. However, with the tests performed, this finding cannot be addressed to a cooling effect of the $CO_2$ shielding gas compared with argon. In future research work at the TU Dresden, the cooling effect of the shield gases will be directly investigated. This knowledge can further be used to develop welding processes with less radiation emission, keeping in mind that health and safety regulations restrict the allowable ozone levels, which are directly linked to arc radiation emission [16].

**Author Contributions:** Data curation, M.L.; Formal analysis, M.L.; Methodology, M.T.; Project administration, M.T.; Resources, U.F.; Supervision, M.T. and U.F.; Validation, M.L., M.T. and S.R.; Visualization, M.L.; Writing—original draft, M.L.; Writing—review & editing, M.L., M.T. and S.R. All authors have read and agreed to the published version of the manuscript.

**Funding:** This research received no external funding.

**Conflicts of Interest:** The authors declare no conflict of interest.

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
