# Peer review of "Influence of the CO2 Content in Shielding Gas on the Temperature of the Shielding Gas Nozzle during GMAW Welding"

_jmmp, doi:10.3390/jmmp4040113_

Round 1
Reviewer 1 Report
The article deals with a new scientific topic. The research carried out by the authors is reliable, but the figures require improvement:
- some data is unreadable, the used text frames cut off markings on fig. 8, 13.
- no subscripts or superscripts on some figures , e.g. there is CO2 should be CO2 (fig.8, 13, 14)
- lack of terminological uniformity (I suggest keeping the same markings, e.g. current instead of strom, eg. Fig 10 or describe the differences between used words “current” and “strom” in text)
Please kindly clarify:
What is the measurement error of the camera used, is it important for the measurement results? (1 sentence is enough).
On the fig 8 d it was presented, that there is 200 % of gas mixture. Please kindly correct this.
The conclusions from the work are quite general. The strength of the article is to outline the direction of future research.
Author Response
|
some data is unreadable, the used text frames cut off markings on fig. 8, 13.
|
done |
|
no subscripts or superscripts on some figures , e.g. there is CO2 should be CO2 (fig.8, 13, 14)
|
Excel does not allow for subscripts in legends, only in titles and axis descriptions. The legend entries will be overlayed with textboces allowing for subscripts |
|
lack of terminological uniformity (I suggest keeping the same markings, e.g. current instead of strom, eg. Fig 10 or describe the differences between used words “current” and “strom” in text) |
"strom" is replaced by "current" |
|
On the fig 8 d it was presented, that there is 200 % of gas mixture. Please kindly correct this. |
done |
|
What is the measurement error of the camera used, is it important for the measurement results? (1 sentence is enough). |
sentence enhanced “The process was recorded with the help of an SLR camera and a high-speed camera in order to assess the visible part of the arc in a qualitative manner.” |
|
The conclusions from the work are quite general. The strength of the article is to outline the direction of future research. |
the conclusion have been extended “In future research work at the TU Dresden the cooling effect of the shield gases will be directly investigated. This knowledge can further be used to develop welding processes with less radiation emission, keeping in mind that health and safety regulations restrict the allowable ozone levels, which are directly linked to arc radiation emission [16].” |
Reviewer 2 Report
Journal: Journal of Manufacturing and Materials Processing
Manuscript ID: jmmp-1015944
Type of manuscript: Article
Title: Influence of the CO2 content in the shielding gas on the temperature
of the shielding gas nozzle during GMAW welding
Authors: Martin Lohse *, Marcus Trautmann, Uwe Füssel, Sascha Rose
In this work the influence of shielding gas composition on welding process is investigated. In particular, the main attention is paid to measuring the radiation part of heat generation by calorimetry. It is shown that in different modes the composition of the gas mixture affects in a complex way. The setting of the experiment itself has novelty and certain interest. However, the practical significance of the results obtained so far seems weak. In general, the work has been performed correctly. There are several remarks to the article:
- The units of measurement in image 1 should be given a standard form. For example, not «J kg^-1 K^-1», but «J/kg·K».
- In Figure 2 you need to fix the axes signatures (as in the other figures).
- In Figure 3 you need to change the units of measurement in the standard form (not exponential).
- Why was the value 300 A selected? To keep the current constant, you have to change the feed rate. This changes the voltage and power. In the end, the energy per meter of weld is different. This causes the purity of the experiment to be disturbed because the full power of the process was changed in each mode. I think it would be much easier to establish the effect of the shielding gas mixture at a constant power.
- Incorrect histogram captions in Figures 8 and 13.
- The radiation part of the heat dissipation is a few percent of the total power. The relevance of measurement of this part is weakly revealed in the work.
- Wire material is not specified.
Author Response
|
The units of measurement in image 1 should be given a standard form. For example, not «J kg^-1 K^-1», but «J/kg·K». |
done |
|
In Figure 2 you need to fix the axes signatures (as in the other figures). |
done |
|
In Figure 3 you need to change the units of measurement in the standard form (not exponential). |
done |
|
Why was the value 300 A selected? To keep the current constant, you have to change the feed rate. This changes the voltage and power. In the end, the energy per meter of weld is different. This causes the purity of the experiment to be disturbed because the full power of the process was changed in each mode. I think it would be much easier to establish the effect of the shielding gas mixture at a constant power. |
an explanation as to why 300A were chosen was added “These parameters were chosen following the AiF project [6], in order to achieve the same welding current for all shielding gases investigated”
In order to look into the influence of the absorbed radiation the radiating volume, i.e. the arc, is of interest. Therefore, the authors tried to maintain a constant radiating volume. this is discussed in the paper |
|
Incorrect histogram captions in Figures 8 and 13. |
captions made readable |
|
The radiation part of the heat dissipation is a few percent of the total power. The relevance of measurement of this part is weakly revealed in the work |
the heat input into the torch can be quite considerable due to radiation explanation added to state of the art “approx. 7-9 % of the electrically supplied power of the welding process is dissipated by cooling the torch. Depending on the process, this can be up to a Kilowatt and explains the need for water-cooling in welding torches under high loads. However, no publication could be found on how large the contribution of heat radiation, especially for different shielding gases is.” |
|
Wire material is not specified |
definition added |
Reviewer 3 Report
The paper is interesting and useful, well-structured and well readable. The authors present influence of the CO2 content in the shielding gas on the temperature of the shielding gas nozzle during GMAW welding. Despite the idea of such a study is not entirely new, the manuscript does contain novel results, especially in heat input measurements. Based on the obtained results, a very detailed description of influence of the shielding gas in GMAW welding on the energy input into the shielding gas nozzle due to radiation was presented. The presented analysis is logical, and the number of references is sufficient. In general, the presented results are of interest as for scientists and engineers. Results of the research are relatively clear, but the manuscript needs a minor revision. The minor comments are given as follows:
Figure 1. „Specific heat”, „thermal conductivity” and its units should be on the left, next to the axis.
Figures 8 and 13. Values of heat input are not visible.
Are the welding parameters based on the own research results or based on parameters described in literature? If parameters were used according to existed literature database, please add a proper reference.
Author Response
|
Figure 1. „Specific heat”, „thermal conductivity” and its units should be on the left, next to the axis.
|
done |
|
Figures 8 and 13. Values of heat input are not visible.
|
done |
|
Are the welding parameters based on the own research results or based on parameters described in literature? If parameters were used according to existed literature database, please add a proper reference.
|
reference added “These parameters were chosen following the AiF project [6], in order to achieve the same welding current for all shielding gases investigated” |